# *BIRC6* modifies risk of invasive bacterial infection in Kenyan children

James J Gilchrist[1,2,3]*, Silvia N Kariuki[4], James A Watson[5,6], Gavin Band[3], Sophie Uyoga[4], Carolyne M Ndila[4], Neema Mturi[4], Salim Mwarumba[4], Shebe Mohammed[4], Moses Mosobo[4], Kaur Alasoo[7], Kirk A Rockett[3], Alexander J Mentzer[3], Dominic P Kwiatkowski[3,8], Adrian VS Hill[3,9], Kathryn Maitland[4,10], J Anthony G Scott[4,11], Thomas N Williams[4,12]*

[1]Department of Paediatrics, University of Oxford, Oxford, United Kingdom; [2]MRC–Weatherall Institute of Molecular Medicine, University of Oxford, Oxford, United Kingdom; [3]Wellcome Centre for Human Genetics, University of Oxford, Oxford, United Kingdom; [4]KEMRI-Wellcome Trust Research Programme, Centre for Geographic Medicine Research-Coast, Kilifi, Kenya; [5]Centre for Tropical Medicine and Global Health, Nuffield Department of Medicine, University of Oxford, Oxford, United Kingdom; [6]Mahidol Oxford Tropical Medicine Research Unit, Faculty of Tropical Medicine, Mahidol University, Bangkok, Thailand; [7]Institute of Computer Science, University of Tartu, Tartu, Estonia; [8]Wellcome Sanger Institute, Cambridge, United Kingdom; [9]The Jenner Institute, University of Oxford, Oxford, United Kingdom; [10]Division of Medicine, Imperial College, London, United Kingdom; [11]Department of Infectious Disease Epidemiology, London School of Hygiene & Tropical Medicine, London, United Kingdom; [12]Institute for Global Health Innovation, Department of Surgery and Cancer, Imperial College, London, United Kingdom

*For correspondence:
james.gilchrist@paediatrics.ox.ac.uk (JJG);
TWilliams@kemri-wellcome.org (TNW)

**Competing interest:** The authors declare that no competing interests exist.

**Abstract** Invasive bacterial disease is a major cause of morbidity and mortality in African children. Despite being caused by diverse pathogens, children with sepsis are clinically indistinguishable from one another. In spite of this, most genetic susceptibility loci for invasive infection that have been discovered to date are pathogen specific and are not therefore suggestive of a shared genetic architecture of bacterial sepsis. Here, we utilise probabilistic diagnostic models to identify children with a high probability of invasive bacterial disease among critically unwell Kenyan children with *Plasmodium falciparum* parasitaemia. We construct a joint dataset including 1445 bacteraemia cases and 1143 severe malaria cases, and population controls, among critically unwell Kenyan children that have previously been genotyped for human genetic variation. Using these data, we perform a cross-trait genome-wide association study of invasive bacterial infection, weighting cases according to their probability of bacterial disease. In doing so, we identify and validate a novel risk locus for invasive infection secondary to multiple bacterial pathogens, that has no apparent effect on malaria risk. The locus identified modifies splicing of *BIRC6* in stimulated monocytes, implicating regulation of apoptosis and autophagy in the pathogenesis of sepsis in Kenyan children.

## Editor's evaluation

Gilchrist et al. present evidence that a genetic variant in the BIRC6 gene increases the risk of invasive bacterial infection. This paper will be of interest to researchers working in areas relating to invasive bacterial infections, malaria, sepsis, and immunogenetics. The authors find evidence that the risk allele affects the splicing of BIRC6 in stimulated monocytes and that this may explain the

association signal they found. Further investigations in African ancestry samples will be needed to elucidate the mechanism.

## Introduction

Invasive bacterial diseases are a major cause of child morbidity and mortality in Africa (*Berkley et al., 2005*). Although improved control measures, including the rollout of anti-bacterial conjugate vaccines (*Cowgill et al., 2006*; *Silaba et al., 2019*), have led to recent declines, the burden of conditions such as pneumonia, meningitis, and sepsis secondary to bacterial pathogens remains significant (*Vos, 2020*). A better understanding of the biology of invasive bacterial infections in African populations might help the development of novel interventions.

Susceptibility to invasive bacterial infections varies widely between individuals. In African children, some of this variability is determined by acquired comorbidities such as HIV, malnutrition, and malaria (*Berkley et al., 2005*; *Church and Maitland, 2014*; *Scott et al., 2011*). However, the identification of common genetic variants as determinants of bacterial infection suggests that a significant portion of this variability is heritable. Many of these genetic susceptibility loci have pathogen-specific effects (*Davila et al., 2010*; *Gilchrist et al., 2018*; *Rautanen et al., 2016*), which is consistent with our understanding of infection susceptibility derived from primary immunodeficiencies. Key examples of pathogen specificity among primary immunodeficiencies include Mendelian susceptibility to myco-bacterial disease and susceptibility to non-tuberculous mycobacteria and nontyphoidal *Salmonella* (*Bustamante et al., 2014*), terminal complement deficiencies and meningococcal disease (*Figueroa et al., 1993*), and IRAK4 deficiency and pneumococcal disease (*Picard et al., 2007*). A major exception to this is the rs334 A>T mutation in *HBB* (sickle haemoglobin), which is associated with raised and lowered risks of infection secondary to a broad range of pathogens among homozygotes (*Williams et al., 2009*) and heterozygotes (*Scott et al., 2011*), respectively. However, the effect sizes associated with sickle haemoglobin are extreme, being maintained in populations by balancing selection, and larger sample sizes will probably be needed for the discovery of new variants with multi-pathogen effects.

Because the clinical features of invasive bacterial infections and severe malaria are broadly similar (*Bejon et al., 2007*), it can be difficult to distinguish between severe illness caused by extensive sequestration of malaria parasites in the microvasculature ('true' severe malaria) and bacterial sepsis in the presence of incidental parasitaemia on the basis of clinical features alone. This is made harder still by the fact that antibiotic pre-treatment and inadequate blood culture volumes mean that, even in settings with excellent diagnostic facilities, true invasive bacterial infections can often not be confirmed (*Driscoll et al., 2017*). Recently, we illustrated this clinical complexity through a study in which we used probabilistic models based on malaria-specific biomarkers to show that approximately one-third of children recruited to studies in Africa with a clinical diagnosis of severe malaria were actually suffering from other conditions (*Watson et al., 2021a*; *Watson et al., 2021b*).

In the current study, we extend this work to show that invasive bacterial infections are common in children admitted to hospital with a clinical diagnosis of severe malaria, but in whom biomarkers subsequently suggest that malaria was probably not the primary cause for their severe illness. We then construct a dataset of genome-wide genotyped samples from 5400 Kenyan children, comprising critically unwell Kenyan children with bacteraemia (*Rautanen et al., 2016*) and severe malaria (*Band, 2019*), and population controls. Using this dataset, we perform a genome-wide association study (GWAS) of invasive bacterial infection in Kenyan children, weighting cases according to the probability that their disease was mediated by a bacterial pathogen. In doing so, we increase our study power and identify *BIRC6* as a novel genetic determinant of invasive bacterial disease in Kenyan children.

## Results

### Severe malaria probability and risk of bacteraemia

Children admitted to the high dependency ward of Kilifi County Hospital with a clinical diagnosis of severe malaria, defined as a severe febrile illness in the presence of *Plasmodium falciparum* parasi-taemia (n=2200), between 11 June 1995 and 12 June 2008 were included in the study. While this definition is sensitive it is not specific, meaning that our study will have included some children with

**eLife digest** Bacterial infections are a major cause of severe illness and death in African children. Understanding which children are at risk of life-threatening infection and why, is key to designing new tools to help protect them. Some risk is likely inherited, but scientists do not know which genes are responsible. Genome-wide association studies (GWAS) may be one way to identify bacterial infection risk genes. GWAS look for genetic differences associated with a particular disease. But previous GWAS studies have failed to find genes linked with bacterial infections in African children because they were too small.

Malaria is another frequent cause of life-threatening illness in African children. It can be hard for clinicians to determine if a child's illness is caused by malaria, a bacterial infection, or both. Many children in Africa have malaria parasites in their blood, but they do not always cause disease. Most children with suspected severe malaria are treated with antibiotics in case of bacterial infection. Clinicians may then conduct further testing to determine the illness's actual cause. Scientists may be able to use this data on children with suspected malaria to study bacterial infections.

Gilchrist et al. show that children with an unusual alteration in the BIRC6 gene are at increased risk of bacterial infections. In the experiments, Gilchrist et al. used computer modeling to identify a subset of children with likely bacterial infections among 2,200 children admitted to a hospital in Kenya with a high fever and malaria parasites. By combining information on this subset of children with data on children with confirmed bacterial infections and healthy children, Gilchrist created a sample of 5,400 children for a GWAS. The analyses found that children with a variation in the BIRC6 gene on chromosome 2 had a higher risk of bacterial infections.

This genetic change is linked with the production of a modified form of BIRC6 in infection-fighting immune cells called monocytes. More studies will help scientists understand how this change might contribute to severe bacterial infections. Learning more may help scientists develop new treatment strategies and identify children most at risk.

sepsis accompanied by incidental parasitaemia (*Watson et al., 2021a*). We therefore used two probabilistic models, which included either platelet counts and plasma *Pf*HRP2 concentrations (Model 1, n=1400) or white blood cell and platelet counts (Model 2, n=2200), to determine the likelihood of 'true' severe malaria among these children. The estimated probabilities of 'true' severe malaria using each model were well correlated (r = 0.64). Of 1400 children with a clinical diagnosis of severe malaria with measured plasma *Pf*HRP2 concentrations, 425 (30.4%, *Figure 1A and B*) had a low probability (P(SM|Data)<0.5) of having 'true' severe malaria (941 of 2220 children using WBC and platelet count data, *Figure 1—figure supplement 1A and B*). That is, while they presented with febrile illness and concomitant malaria parasitaemia, it is unlikely that their illnesses were directly attributable to malaria.

In keeping with the hypothesis that a significant proportion of these critically unwell children represented culture-negative invasive bacterial disease (*Figure 1*), in-patient mortality was higher among children with a low than a high probability of 'true' severe malaria (*Table 1*; $\text{OR}_{\text{model1}}$ = 1.57, 95% CI $1.11 - 2.21$, $p = 0.01$, 95% CI $1.60 - 2.72$, $p = 4.91 \times 10^{-8}$). This was also reflected in the rates of concurrent bacteraemia (*Table 1*; $\text{OR}_{\text{model1}}$ = 2.92, 95% CI $1.66 - 5.13$, $p = 1.07 \times 10^{-4}$, 95% CI $1.27 - 3.17$, $p = 0.003$). Similarly, the constituents of Model 1 were each associated with blood culture positivity, both higher platelet counts (OR = 2.36, 95% CI 1.19–4.70, $p = 0.014$) and lower *Pf*HRP2 levels (OR = $0.52, 95\%$ CI $0.390.70$, $p = 9.62 \times 10^{-6}$) both being associated with the risk of coincident bacteraemia (*Figure 1C and D*). Conversely, white blood counts in isolation were not associated with risk of concurrent bacteraemia (*Figure 1—figure supplement 1*). Plasma *Pf*HRP2 is the single best biomarker for severe malaria (*Hendriksen et al., 2012*). In light of this, and given the greater enrichment for concurrent bacteraemia among children with a low probability of 'true' severe malaria as calculated by Model 1 than Model 2, we used Model 1 probabilities in downstream analyses where available (n=1400) and used Model 2 probabilities for all other cases (n=800).

## GWAS of invasive bacterial disease in Kenyan children

Children with a clinical diagnosis of severe malaria but a low probability of having 'true' severe malaria are thus enriched for invasive bacterial disease. Using genome-wide genotyping data from cases of

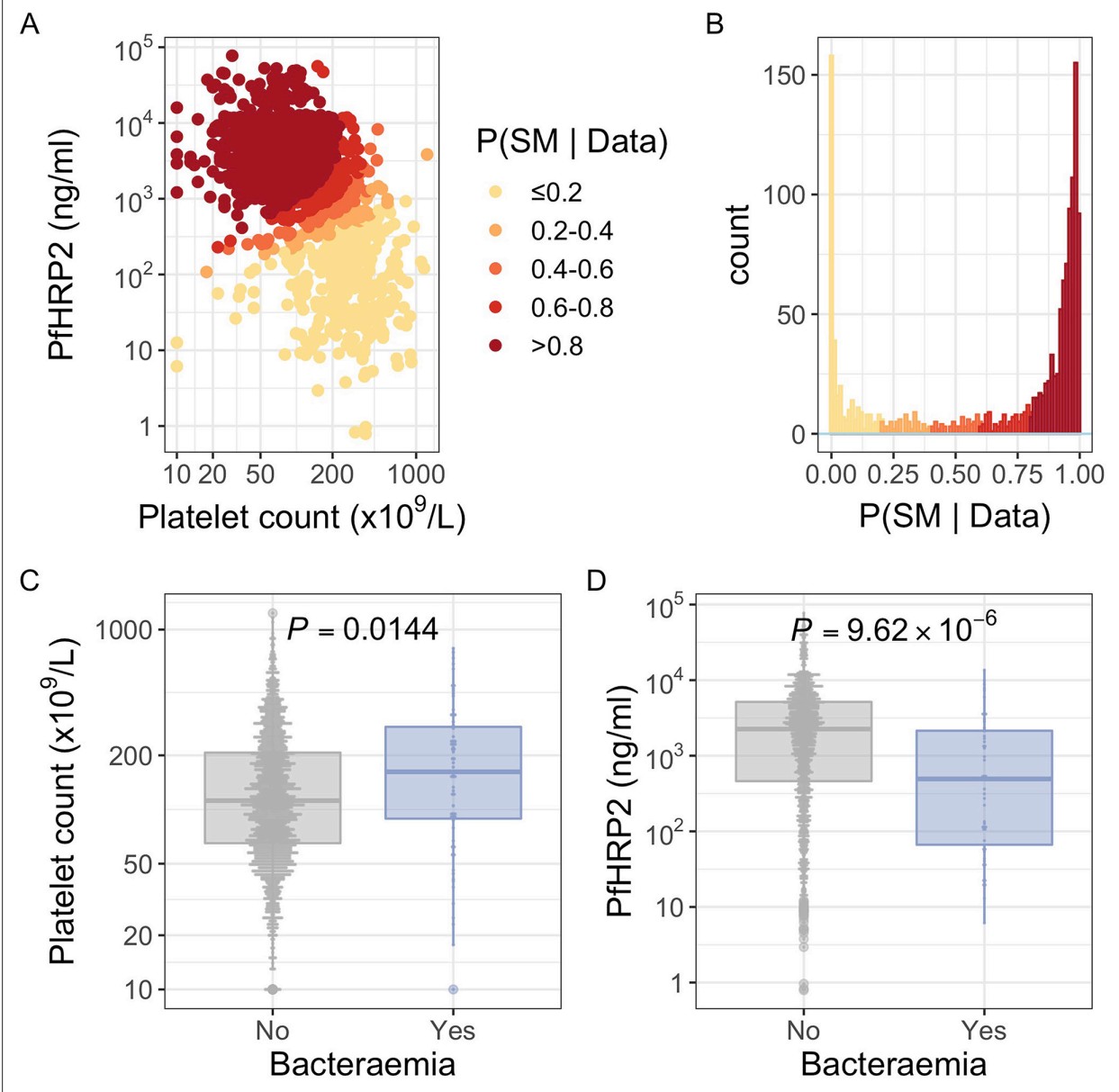

**Figure 1.** *Pf*HRP2 and platelet count as predictors of severe malaria. (**A**) Distribution of *Pf*HRP2 concentrations and platelet count among Kenyan children (n=1400) with a clinical diagnosis of severe malaria. Points are coloured according to the probability of 'true' severe malaria given the data. (**B**) Distribution of 'true' severe malaria probabilities estimated from platelet count and plasma *Pf*HRP2 concentrations. (**C**) Platelets counts in children with a clinical diagnosis of severe malaria with and without concomitant bacteraemia. (**D**) *Pf*HRP2 concentrations in children with a clinical diagnosis of severe malaria with and without concomitant bacteraemia. Box and whisker plots; boxes depict the upper and lower quartiles of the data, and whiskers depict the range of the data excluding outliers (outliers are defined as data-points > 1.5 × the inter-quartile range from the upper or lower quartiles). Comparisons of platelet count and plasma *Pf*HRP2 concentrations in children with and without bacteraemia were performed with logistic regression.

The online version of this article includes the following figure supplement(s) for figure 1:

**Figure supplement 1.** White blood cell and platelet count as predictors of severe malaria.

culture-confirmed bacteraemia and healthy controls, we estimated SNP heritability of bacteraemia in this population to be 19% (95% CI 3–35%, $p = 0.0084$). Despite this, our GWAS of bacteraemia failed to identify genetic correlates of bacteraemia risk beyond the sickle cell locus (*Rautanen et al., 2016*). Motivated by these observations, we performed a GWAS of invasive bacterial infection in Kenyan children in which we included both children with culture-confirmed bacteraemia and children with a clinical diagnosis of severe malaria. Children admitted to Kilifi County Hospital between 1 August

**Table 1.** Demographics and clinical characteristics of Kenyan children with severe malaria.

| Model | Numbers | Sex (female) | Age (months) | Bacteraemia | Mortality |
|---|---|---|---|---|---|
| | Total (n=1400) | 695 (49.6%) | 29 (17–44) | 51 (3.6%) | 155 (11.1%) |
| | P(SM\|Data)>0.5 (n=975) | 497 (51.0%) | 29 (18–45) | 23 (2.4%) | 94 (7.4%) |
| *Pf*HRP2/Plt | P(SM\|Data)<0.5 (n=425) | 198 (46.6%) | 28 (16–43) | 28 (6.6%) | 61 (14.4%) |
| | Total (n=2220) | 1,074 (48.4%) | 28 (15–43) | 78 (3.5%) | 256 (11.6%) |
| | P(SM\|Data)>0.5 (n=1279) | 623 (48.7%) | 29 (17–44) | 32 (2.5%) | 106 (8.4%) |
| WBC/Plt | P(SM\|Data)<0.5 (n=941) | 451 (47.9%) | 25 (13–40) | 46 (4.9%) | 150 (15.9%) |

Mortality reflects in-patient deaths. Figures are absolute numbers with percentages or interquartile ranges in parentheses. P(SM|Data) reflects the probability of 'true' severe malaria estimated from each model (*Pf*HRP2/Platelet count, White blood cell count/Platelet count).

1998 and 30 October 2010 with community-acquired bacteraemia were recruited to the study as well as children from the severe malaria study described above. Control children were recruited from the same population between 1 August 2006 and 30 December 2010 as described in detail previously (*Scott et al., 2011*).

Following quality control measures (see Materials and methods), we included 1445 cases of culture-confirmed bacteraemia, 1143 cases of severe malaria, and 2812 control children in our current analysis (*Table 2*, *Figure 2*). To account for the varying proportion of invasive bacterial disease among severe malaria cases, we applied weights to our regression analysis to reflect the greater likelihood of invasive bacterial disease among children with a low probability of 'true' severe malaria (sample weight, w = 1 − P(SM|Data)). Where *Pf*HRP2 concentrations were available (n=909) we used *Pf*HRP2 and platelet count to determine P(SM|Data) while we used white cell and platelet counts (n=234) in cases where they were not available. Cases with culture-proven bacteraemia and control samples were assigned a sample weight of *w* = 1. Inclusion of the six major principal components

**Table 2.** Demographics and clinical characteristics of genome-wide association study (GWAS) study samples.

| | Numbers | Sex (female) | Age (months) | Severe malaria subtypes | | Concurrent infection | | Mortality |
|---|---|---|---|---|---|---|---|---|
| | | | | SMA | CM | Malaria | Bacteraemia | |
| Bacteraemia (overall) | 1445 | 614 (43%) | 14 (5–34) | | | 94 (12%) | | 358 (26%) |
| *Acinetobacter* | 118 | 45 (38%) | 13 (3–28) | | | 11 (13%) | | 12 (10%) |
| β-Haemolytic streptococci | 130 | 60 (46%) | 5 (1–20) | | | 6 (8%) | | 37 (30%) |
| *Escherichia coli* | 141 | 58 (41%) | 11 (6–25) | | | 12 (15%) | | 45 (34%) |
| Hib | 113 | 53 (47%) | 12 (5–25) | | | 3 (8%) | | 29 (26%) |
| NTS | 159 | 75 (47%) | 15 (9–26) | | | 15 (25%) | | 31 (20%) |
| *Streptococcus pneumoniae* | 390 | 151 (39%) | 23 (9–61) | | | 20 (9%) | | 86 (23%) |
| *Staphylococcus aureus* | 152 | 64 (42.1%) | 26 (9–88) | | | 15 (15%) | | 22 (15%) |
| Other | 242 | 110 (46%) | 10 (1–28) | | | 10 (8%) | | 96 (41%) |
| Malaria (overall) | 1143 | 559 (49%) | 27 (16–41) | 298 (26%) | 697 (61%) | | 40 (4%) | 140 (12%) |
| P(SM\|Data)<0.5 | 375 | 176 (47%) | 28 (17–42) | 62 (17%) | 262 (70%) | | 23 (6%) | 62 (17%) |
| P(SM\|Data)>0.5 | 768 | 383 (50%) | 26 (15–40) | 236 (31%) | 435 (57%) | | 17 (2%) | 17 (2%) |
| Controls (SNP 6.0) | 1895 | 955 (50%) | | | | | | |
| Controls (Omni 2.5M) | 917 | 451 (49.2%) | * | | | | | |

P(SM|Data) reflects the probability of 'true' severe malaria estimated from *Pf*HRP2 concentration and platelet count or white blood cell count and platelet count. Blood cultures were taken from all children severe malaria at admission. *Control children were recruited between 3 and 12 months of age and have been subject to longitudinal follow-up. SMA, severe malarial anaemia; CM, cerebral malaria.

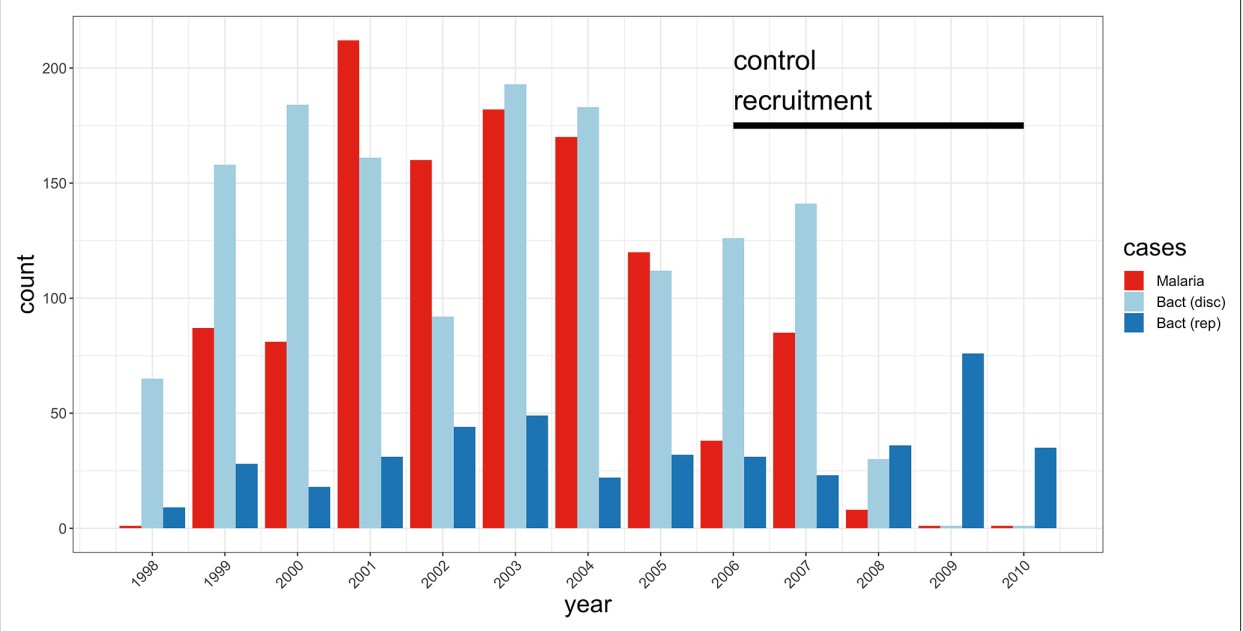

**Figure 2.** Study sample recruitment. Recruitment of severe malaria cases (n=1143), discovery analysis bacteraemia cases (n=1445), and replication analysis bacteraemia cases (n=434) between 1998 and 2010. The time period during which control samples (n=2812) were recruited is also highlighted (black bar). Sample numbers represent children with genome-wide genotype data, who passed quality control filters and were included in this study.

(PCs) of genotyping data and genotyping platform as covariates in the model adequately controlled for confounding variation ($\lambda = 1.0208$, **Figure 3—figure supplement 1**). In that analysis we found evidence supporting an association between risk of invasive bacterial disease in Kenyan children and seven SNPs at a single locus on chromosome 2 (peak SNP: rs183868412:T, OR = 2.13, 95% CI 1.65–2.74, $p = 4.64 \times 10^{-9}$) (**Figure 3**, **Table 3**). Fine mapping of this association identified a credible set of seven SNPs with a 95% probability of containing the causal variant (**Table 3**), spanning a 212 kb region: chr2:32,402,640–32,614,746.

To address the possibility that the observed association at this locus is driven by confounding secondary to population structure, we used ABERRANT (**Bellenguez et al., 2012**) to define a set of outlier samples on the first two PCs of genotyping data (n=22, **Figure 3—figure supplement 2**). These individuals are all genotyped on the Illumina Omni 2.5M array and are both cases and controls

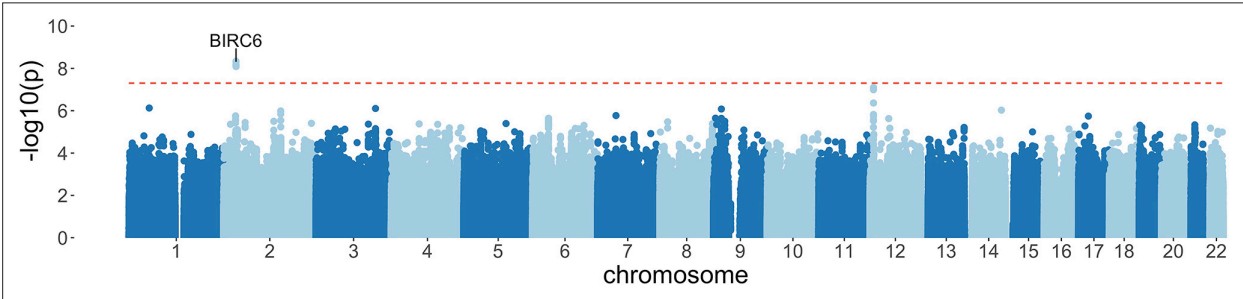

**Figure 3.** Manhattan plot of invasive bacterial infection in Kenyan children. Evidence for association with invasive bacterial disease at genotyped and imputed autosomal SNPs and indels (n=14,010,600) in Kenyan children (bacteraemia cases =1445, severe malaria cases =1143, controls = 2812). Association statistics represent additive association. The red, dashed line denotes $p = 5 \times 10^{-8}$.

The online version of this article includes the following figure supplement(s) for figure 3:

**Figure supplement 1.** Genome-wide association analysis quality control.

**Figure supplement 2.** Principal components of genome-wide genotyping data in discovery samples.

**Figure supplement 3.** Sensitivity analysis of case weights.

**Figure supplement 4.** Genotyping concordance between Illumina and Affymetrix platforms.

**Table 3.** 95% credible SNP set of invasive bacterial disease association.

| SNP | Effect allele | Chr | BP | MAF | Info score | OR (95% CI) | p-Value |
|---|---|---|---|---|---|---|---|
| rs183868412 | T | 2 | 32,478,169 | 0.021 | 0.956 | 2.13 (1.65–2.74) | $4.64 \times 10^{-9}$ |
| rs139827594 | G | 2 | 32,402,640 | 0.020 | 0.966 | 2.12 (1.65–2.73) | $4.96 \times 10^{-9}$ |
| rs144257579 | G | 2 | 32,507,619 | 0.021 | 0.954 | 2.11 (1.64–2.72) | $6.82 \times 10^{-9}$ |
| rs145056232 | C | 2 | 32,503,024 | 0.021 | 0.955 | 2.11 (1.64–2.72) | $6.86 \times 10^{-9}$ |
| rs145315025 | G | 2 | 32,502,654 | 0.021 | 0.955 | 2.11 (1.64–2.72) | $6.87 \times 10^{-9}$ |
| rs143909151 | T | 2 | 32,531,452 | 0.021 | 0.962 | 2.11 (1.64–2.71) | $8.01 \times 10^{-9}$ |
| rs150430979 | T | 2 | 32,614,746 | 0.021 | 0.955 | 2.11 (1.64–2.72) | $8.18 \times 10^{-9}$ |

MAF = minor allele frequency. CI = confidence interval. Genomic coordinates are GRCh38.

(15 and 7, respectively). While they are predominantly individuals with less common self-reported ethnicities in our study population (19 of 22 are not Giriama, Chonyi, or Kauma), they are not representative of a single self-reported ethnicity (the most common single ethnicity in this group is Digo, n=7). Excluding these samples from the association analysis at rs183868412:T did not significantly alter the association with invasive bacterial infection ($p = 2.38 \times 10^{-8}$, OR = 2.05, 95% CI 1.59–2.64). We further estimated the effect of rs183868412:T on invasive bacterial disease risk in four subpopulations defined by self-reported ethnicity (Giriama, $n = 2,501$; Chonyi, $n = 1,560$; Kauma, $n = 472$; Other, $n = 384$). Within each subpopulation, we tested for association between genotype and case status in a weighted logistic regression model, including platform as a categorial covariate (**Table 4**). The minor allele frequency (MAF) at rs183868412 ranged from 0.016 (Giriama) to 0.037 (Kauma), with no evidence of differentiation between subpopulations ($F_{ST} = 0.001$). We observed consistent effect sizes in both of the major study subpopulations; Giriama (OR =1.97, 95% CI 1.30–3.01, $p = 0.0015$) and Chonyi (OR = 2.18, 95% CI 1.34–3.54, $p = 0.0017$) samples (**Table 4**), which together make up 83% of the study samples. Genotype at rs183868412 was also associated with invasive bacterial disease among the less common self-reported ethnicities grouped together (OR = 2.46, 95% CI 1.01–5.96, $p = 0.047$). Genotype was not associated with invasive bacterial disease risk in the Kauma subpopulation, however the sample size in the stratum is very limited (154 cases, 318 controls) and may simply reflect insufficient power to detect an association.

To assess whether our analysis could be affected by our choice of model to define severe malaria case weights, we restricted our analysis to samples with data available to calculate estimates for $P(SM|Data)$ using both Model 1 and Model 2 (n=909 severe malaria cases). We recalculated effect estimates for the rs183868412 association with invasive bacterial disease using each model alone. The association with invasive bacterial disease at rs183868412:T is robust to the choice of the model for case weights, with effect estimates derived using Model 1 alone (OR = 2.13, 95% CI 1.65–2.75, $p = 6.92 \times 10^{-9}$) and Model 2 alone (OR = 2.05, 95% CI 1.59–2.65, $p = 2.63 \times 10^{-8}$) being entirely consistent with those seen in the main analysis (**Figure 3—figure supplement 3**). Moreover, restricting our analysis to cases of culture-confirmed bacteraemia, the effect estimate for bacteraemia risk observed in the discovery analysis (1445 cases, 2812 controls; OR = 2.12, 95% CI 1.60–2.82, $p = 1.97 \times 10^{-7}$) is consistent with that seen in the main model.

We sought to replicate evidence of association in our discovery analysis through use of an independent case-control collection of Kenyan children with bacteraemia (n=434) and healthy controls (n=1258) conducted in the same population. The peak trait-associated variants in the discovery analysis were well imputed in the replication data (rs183868412 imputation info score = 0.84). In that analysis, we found evidence supporting the association at chromosome 2 with invasive bacterial disease (**Figure 4**, **Table 5**: rs183868412:T, OR = 2.85, 95% CI 1.54–5.28, $p = 8.90 \times 10^{-4}$). In a fixed effects meta-analysis, rs183868412:T was strongly associated with risk of invasive bacterial disease in Kenyan children: OR = 2.22, 95% CI 1.76–2.80, $p = 2.35 \times 10^{-11}$. That association was driven by children with culture-confirmed bacteraemia and critically unwell children with malaria parasites, but a low probability of 'true' severe malaria. In a stratified analysis (**Figure 4**, **Table 5**), rs183868412 was associated

**Table 4.** Effect of rs183868412 genotype on risk of invasive bacterial disease stratifies by self-reported ethnicity.

| Discovery population | | | Numbers | Genotypes | MAF | OR (95% CI) | p-Value |
|---|---|---|---|---|---|---|---|
| | | Overall | 1232 | 0/56/1176 | 0.023 | | |
| | | Bacteraemia | 558 | 0/38/520 | 0.034 | | |
| | | SM − P(SM\|Data)<0.5 | 199 | 0/12/187 | 0.030 | | |
| | Cases | SM − P(SM\|Data)>0.5 | 475 | 0/6/469 | 0.006 | | |
| Giriama | Controls | | 1269 | 0/41/1228 | 0.016 | 1.97 (1.30–3.01) | $p = 0.0015$ |
| | | Overall | 503 | 0/38/465 | 0.038 | | |
| | | Bacteraemia | 238 | 0/27/211 | 0.057 | | |
| | | SM − P(SM\|Data)<0.5 | 105 | 0/4/101 | 0.019 | | |
| | Cases | SM − P(SM\|Data)>0.5 | 160 | 0/7/153 | 0.022 | | |
| Chonyi | Controls | | 1057 | 0/43/1014 | 0.020 | 2.18 (1.34–3.54) | $p = 0.0017$ |
| | | Overall | 154 | 0/8/146 | 0.026 | | |
| | | Bacteraemia | 70 | 0/6/64 | 0.043 | | |
| | | SM − P(SM\|Data)<0.5 | 25 | 0/1/24 | 0.020 | | |
| | Cases | SM − P(SM\|Data)>0.5 | 59 | 0/1/58 | 0.008 | | |
| Kauma | Controls | | 318 | 0/20/298 | 0.031 | 1.20 (0.50–2.85) | $p = 0.686$ |
| | | Overall | 219 | 1/16/202 | 0.041 | | |
| | | Bacteraemia | 101 | 1/8/92 | 0.050 | | |
| | | SM − P(SM\|Data)<0.5 | 38 | 0/5/33 | 0.066 | | |
| | Cases | SM − P(SM\|Data)>0.5 | 80 | 0/3/77 | 0.019 | | |
| Other | Controls | | 165 | 0/7/158 | 0.021 | 2.46 (1.01–5.96) | $p = 0.047$ |
| | | Overall | 2588 | 3/164/2421 | 0.033 | | |
| | | Bacteraemia | 1445 | 3/125/1317 | 0.045 | | |
| | | SM − P(SM\|Data)<0.5 | 375 | 0/20/355 | 0.027 | | |
| | Cases | SM − P(SM\|Data)>0.5 | 768 | 0/19/749 | 0.012 | | |
| Total | Controls | | 2812 | 0/111/2701 | 0.020 | 2.13 (1.65–2.74) | $p = 4.64 \times 10^{-9}$ |

Self-reported ethnicity data is missing in 482 samples (480 of which are cases). Effect estimates derived with weighted logistic regression. p(SM|Data) represent the probability of 'true' severe malaria estimated from plasma *Pf*HRP2 concentration and platelet count (n=909) or white blood cell count and platelet count (n=234). OR, odds ratio. MAF, minor allele frequency. CI, confidence interval.

with culture-confirmed bacteraemia (OR = 2.12, 95% CI 1.60–2.82, $p = 1.94 \times 10^{-7}$) and critical illness with parasitaemia and with a low probability of 'true' severe malaria (P(SM|Data)<0.5: OR = 2.37, 95% CI 1.27–4.43, $p = 6.82 \times 10^{-3}$), but was not associated with risk of critical illness with a high probability of 'true' severe malaria (P(SM|Data)>0.5: $p = 0.823$).

## rs183868412 is associated with risk of invasive bacterial disease secondary to diverse pathogens and is independent of malaria

Previous data describing the genetic risk of invasive bacterial disease in this population have identified pathogen-specific effects. To better-understand the range of pathogens to which genetic variation at *BIRC6* modifies risk, we estimated the effect of rs183868412 on the risk of bacteraemia caused by the seven most common causative pathogens within this population (*Figure 5A*). In that analysis, the data best-supported a model in which genotype increases risk of bacteraemia caused by a broad range of pathogens, including bacteraemia secondary to pneumococcus, nontyphoidal *Salmonellae*, *Escherichia coli*, β-haemolytic streptococci, *Staphylococcus aureus*, and other less common pathogens

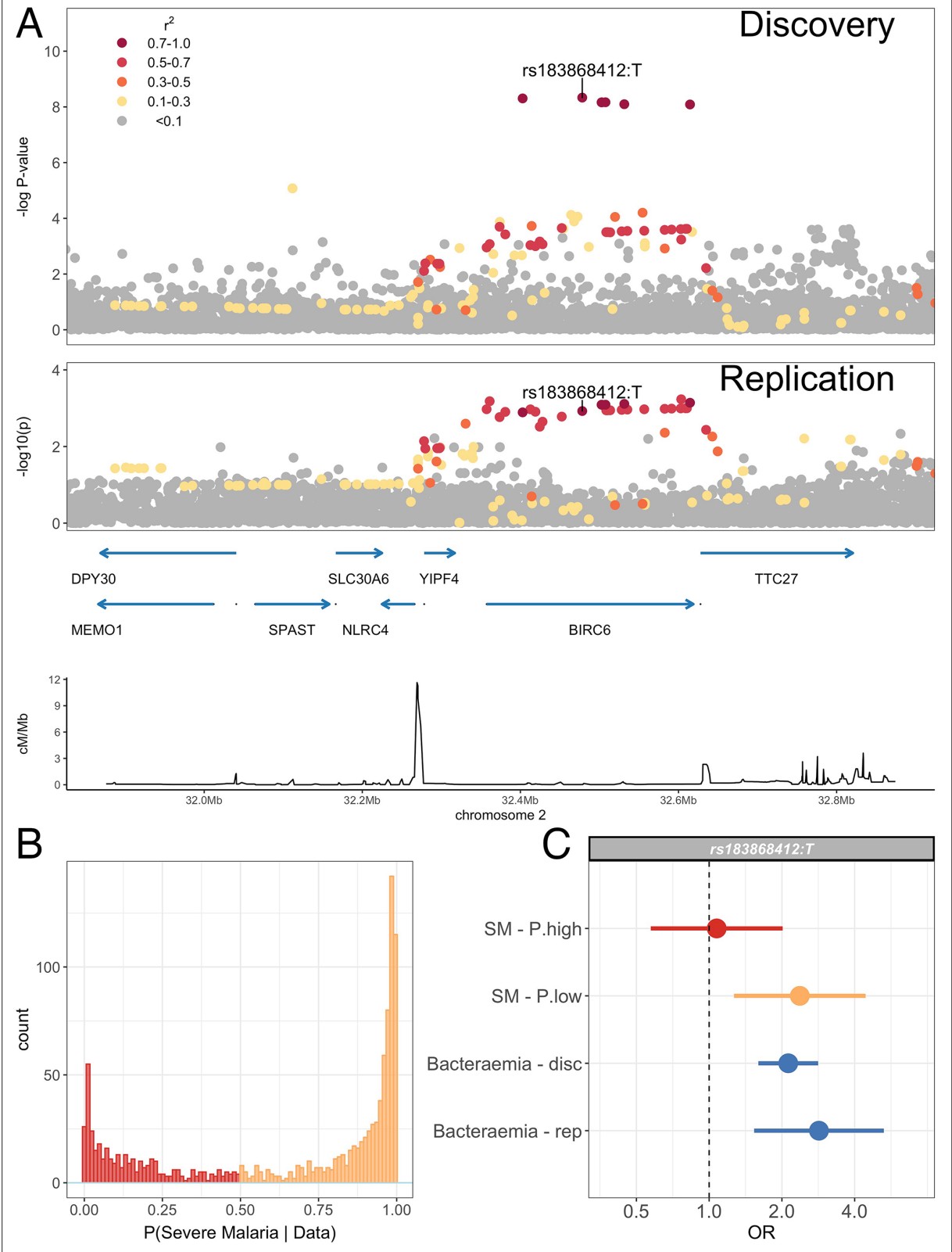

**Figure 4.** Association with invasive bacterial disease at the *BIRC6* locus. (**A**) Regional association plot of invasive bacterial disease association at the *BIRC6* locus in the discovery and replication analyses. SNPs are coloured according to linkage disequilibrium to rs183868412. Genomic coordinates are GRCh38. (**B**) Distribution of 'true' severe malaria probabilities among malaria cases estimated from plasma *Pf*HRP2 concentration and platelet count (n=909) and white blood cell count and platelet count (n=234). (**C**) Odds ratios and 95% confidence intervals of rs183868412 association with disease

*Figure 4 continued on next page*

*Figure 4 continued*

stratified by malaria cases with high (p>0.5, red) and low (p<0.5, orange) probabilities of 'true' severe malaria and culture-proven invasive bacterial disease (blue). P(SM|Data) represents the probability of 'true' severe malaria estimated from plasma *Pf*HRP2 concentration and platelet count (n=909) or white blood cell count and platelet count (n=234). Effect estimates were calculated with multinomial logisitic regression (SM - P.high, SM - P.low, Bacteraemia - disc) and logistic regression (Bacteraemia - rep).

The online version of this article includes the following figure supplement(s) for figure 4:

**Figure supplement 1.** Principal components of genome-wide genotyping data in replication samples.

grouped as a single stratum (log10 Bayes factor = 4.72). Genotype at rs183868412 similarly modified risk of bacteraemia in the neonatal period and in older children (log10 Bayes factor = 2.70, *Figure 5B*).

Malaria infection results in increased risk of invasive bacterial disease secondary to a broad range of pathogens (*Scott et al., 2011*), and genetic determinants of malaria risk (e.g. sickle cell trait) modify risk of bacterial infection in malaria-endemic settings (*Scott et al., 2011*). The observation that, among children with a clinical diagnosis of severe malaria, risk of disease is only modified by rs183868412 in children with a low probability that their disease represents 'true' severe malaria (*Figure 5*) suggests that the effect of genetic variation at *BIRC6* on invasive bacterial disease risk operates independently of malaria. In keeping with this, the data best-supports an effect of rs183868412 genotype on bacteraemia risk in children both with and without concomitant parasitaemia (log10 Bayes factor = 2.73, *Figure 5C*). In addition, unlike sickle cell trait (*Scott et al., 2011*), the increased risk of invasive bacterial infection conferred by rs183868412:T carriage in the study setting is stable across a period of declining malaria prevalence (Bayes factor = 8.7, *Figure 5D*).

## Evidence of selection pressure and pleiotropy at rs183868412

Common genetic variation associated with a twofold increased risk of invasive bacterial infection in children, in particular across a broad range of pathogens, will be subject to considerable negative selection pressure. The derived allele rs183868412:T, associated with increased risk of invasive bacterial disease in Kenyan children, is absent in non-African populations (https://gnomad.broadinstitute.org). Within Africa, rs183868412:T is present in all nine African populations included in the MalariaGEN consortium project (*Band, 2019*; *Table 6*), MAFs ranging from 0.011 in The Gambia to 0.034 in Malawi. There is no evidence for within-Africa differentiation at rs183868412 ($p = 0.601$) providing no support for a selective sweep at the locus. We further evaluated evidence for recent directional selection pressure, examining integrated haplotype scores (iHS) (*Voight et al., 2006*) in seven African populations included in the 1000 Genomes Project. In those data, there is no evidence to support recent selection at the locus (minimum rank $p = 0.07$, maximum iHS = 1.3). Finally, to understand whether

**Table 5.** Effect of rs183868412 genotype on risk of invasive bacterial disease in Kenyan children.

| | | | Numbers | Genotypes | MAF | OR (95% CI) | p-Value |
|---|---|---|---|---|---|---|---|
| | | Overall | 2588 | 3/164/2421 | 0.033 | 2.13 (1.65–2.74) | $p = 4.64 \times 10^{-9}$ |
| | | Bacteraemia* | 1445 | 3/125/1317 | 0.045 | 2.12 (1.60–2.82) | $p = 1.94 \times 10^{-7}$ |
| | | SM − P(SM\|Data)<0.5* | 375 | 0/20/355 | 0.027 | 2.37 (1.27–4.43) | $p = 6.82 \times 10^{-3}$ |
| | Cases | SM − P(SM\|Data)>0.5* | 768 | 0/19/749 | 0.012 | 1.07 (0.57–2.01) | $p = 0.823$ |
| Discovery | Controls | | 2812 | 0/111/2701 | 0.020 | | |
| | Cases | | 434 | 0/24/410 | 0.028 | 2.85 (1.54–5.28) | $p = 8.90 \times 10^{-4}$ |
| Replication | Controls | | 1258 | 0/28/1230 | 0.011 | | |
| | Cases | | 3022 | 3/188/2831 | 0.032 | 2.23 (1.76–2.80) | $p = 2.35 \times 10^{-11}$ |
| Meta-analysis | Controls | | 4070 | 0/139/3931 | 0.017 | | |

*Estimates derived from multinomial logistic regression model. P(SM|Data) represent the probability of 'true' severe malaria estimated from plasma *Pf*HRP2 concentration and platelet count (n=909) or white blood cell count and platelet count (n=234). SM, severe malaria. MAF, minor allele frequency. CI, confidence interval.

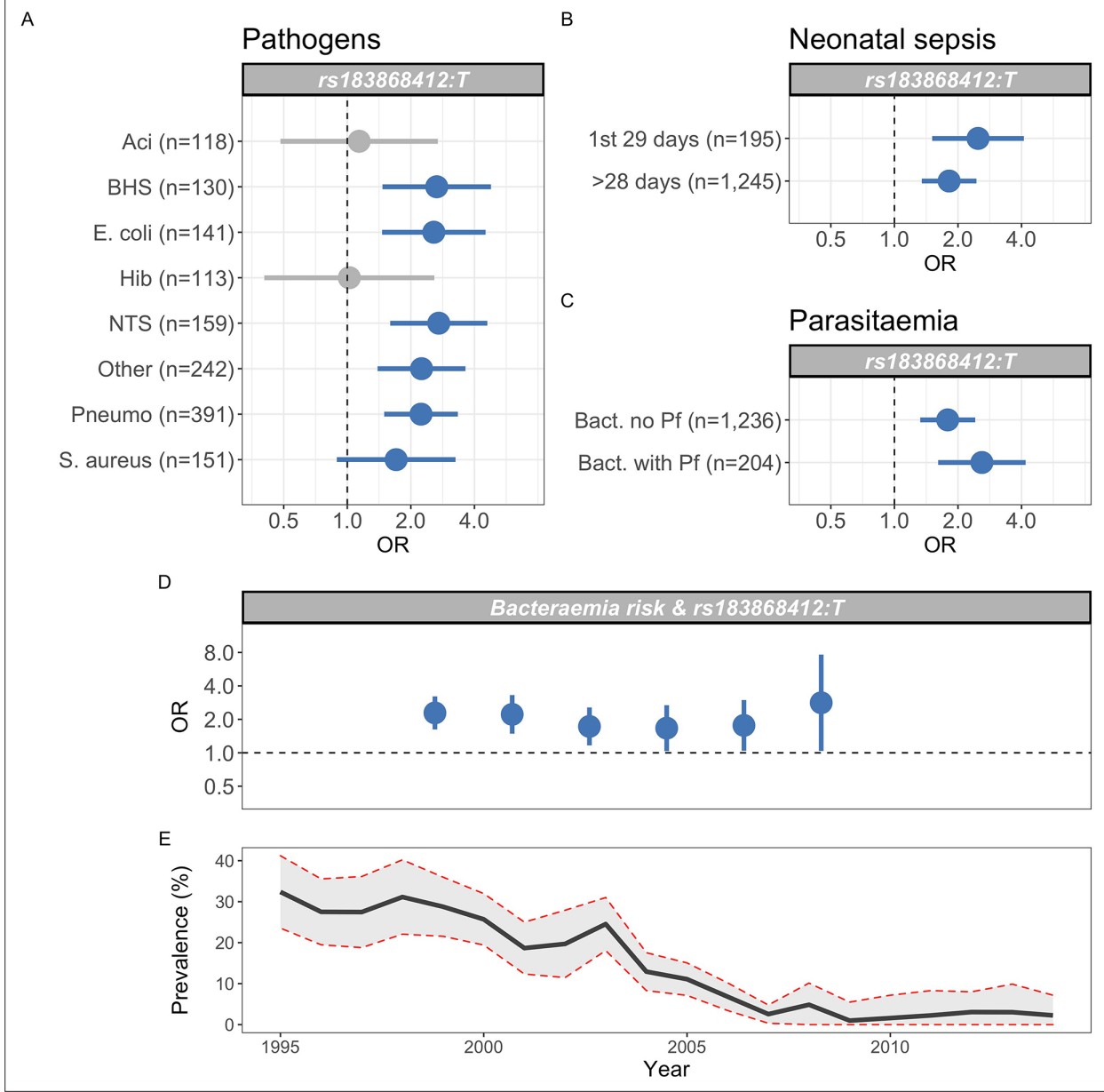

**Figure 5.** Genetic variation at *BIRC6* confers broad susceptibility to invasive bacterial disease. Odds ratios and 95% confidence intervals of rs183868412 association with invasive bacterial disease stratified by pathogen (**A**), neonatal and non-neonatal sepsis (**B**), and bacteraemia with and without malaria parasitaemia (**C**). Odds ratios and 95% confidence intervals of rs183868412 association with invasive bacterial disease stratified by year (**D**), compared to age-standardised, annual malaria parasite prevalence in Kilifi, Kenya, as estimated from parasite prevalence among trauma admissions (**E**).We calculated effect estimates using multinomial logistic regression. We compared models of association across strata using a Bayesian approach (see Materials and methods). Strata associated with rs183868412 genotype in the most likely model in each analysis are highlighted in blue.

variation at this locus may be maintained in populations through beneficial effects on other traits, we examined evidence for pleiotropy at rs183868412 in a meta-analysis of GWAS data (*Gurdasani et al., 2019*) of 33 cardio-metabolic traits from Uganda, Ghana, Nigeria, South Africa, and Kenya (sample size ranging from 2671 to 14,126 individuals). In keeping with the MalariaGEN consortium project data, MAFs at rs183868412 range from 0.015 to 0.028. In these data there is no evidence for pleiotropy at the locus, with no evidence for an effect of rs183868412 on any of the 33 traits tested (minimum meta-analysis $p = 0.078$). The explanation for the persistence of this polymorphism, therefore, remains an open question.

**Table 6.** rs183868412 frequencies in Africa.

| Population | Number | MAF |
|---|---|---|
| Gambia | 2605 | 0.011 |
| Mali | 183 | 0.021 |
| Burkina Faso | 596 | 0.009 |
| Ghana | 320 | 0.014 |
| Nigeria | 22 | 0.024 |
| Cameroon | 685 | 0.031 |
| Malawi | 1317 | 0.034 |
| Tanzania | 402 | 0.028 |
| Kenya | 1681 | 0.017 |

Numbers reflect healthy control samples. MAF, minor allele frequency.

## rs183868412 is associated with alternative splicing of *BIRC6* in stimulated monocytes

Trait-associated genetic variation identified by GWAS is highly enriched for regulatory variation. The African specificity of the trait-associated variation identified here makes annotation with publicly available regulatory mapping data challenging. To investigate the regulatory function of rs183868412 in immune cells in African populations, we used eQTL Catalogue (https://www.ebi.ac.uk/eqtl) mapping data (*Kerimov et al., 2021*) from 100 African ancestry individuals in primary monocytes with and without innate stimulation (*Quach et al., 2016*); influenza A virus, LPS, Pam3CSK4 (synthetic lipoprotein and TLR1/2 agonist) and R848 (a TLR7/8 agonist). In this dataset, rs183868412 is well imputed ($r^2 = 0.998$), with an MAF of 0.05 (10 heterozygous individuals). We found no evidence for a regulatory effect of rs183868412 at the gene level in monocytes regardless of stimulation. We did, however, observe an effect of rs183868412 genotype on expression of a 12 bp *BIRC6* exonic sequence (chr2:32,453,943–32,453,954, $p = 1.18 \times 10^{-5}$), with evidence for colocalisation of this eQTL with our GWAS signal (posterior probability of colocalisation, PP4 = 0.951, *Figure 6*). This effect was only observed following stimulation with Pam3CSK4 (*Figure 6*), with the bacteraemia risk allele, rs183868412:T, being associated with reduced expression of this sequence. That 12 bp sequence is associated with an alternative splicing event that results in extension of a *BIRC6* exon. The 23rd exon (ENSE00001189810, chr2:32,453,808–32,453,942) of the canonical *BIRC6* transcript, ENST00000421745.6, is 135 bp long and terminates immediately before the 12 bp sequence associated with rs183868412:T genotype. The 22nd exon (ENSE00003835010, chr2:32,453,808–32,453,942) of an alternative *BIRC6* transcript, ENST00000648282.1, is 147 bp long, having the same start site but including the 12 bp sequence at its 3' end. Thus, increased risk of invasive bacterial disease may be associated with decreased expression of an alternative *BIRC6* transcript in TLR1/2-stimulated monocytes.

## Discussion

In this study, we have leveraged the close relationship between *P. falciparum* infection and bacteraemia in African children (*Scott et al., 2011*) to perform a GWAS of invasive bacterial infection in 5400 Kenyan children. We approached this by defining the probability with which each critically unwell child with a clinical diagnosis of severe malaria has a disease process directly mediated by malaria, that is 'true' severe malaria. We hypothesised that critically unwell children, with a low probability of having 'true' severe malaria, are enriched for invasive bacterial infections. We explored the validity of this approach, demonstrating that children with a low probability of 'true' severe malaria were indeed enriched for culture-proven bacteraemia and were at a higher risk of death than children with a higher probability. We therefore performed a GWAS weighting cases according to their likelihood of invasive bacterial disease. In doing so, we have identified and validated *BIRC6* as a novel genetic susceptibility locus for all-cause invasive bacterial disease in Kenyan children.

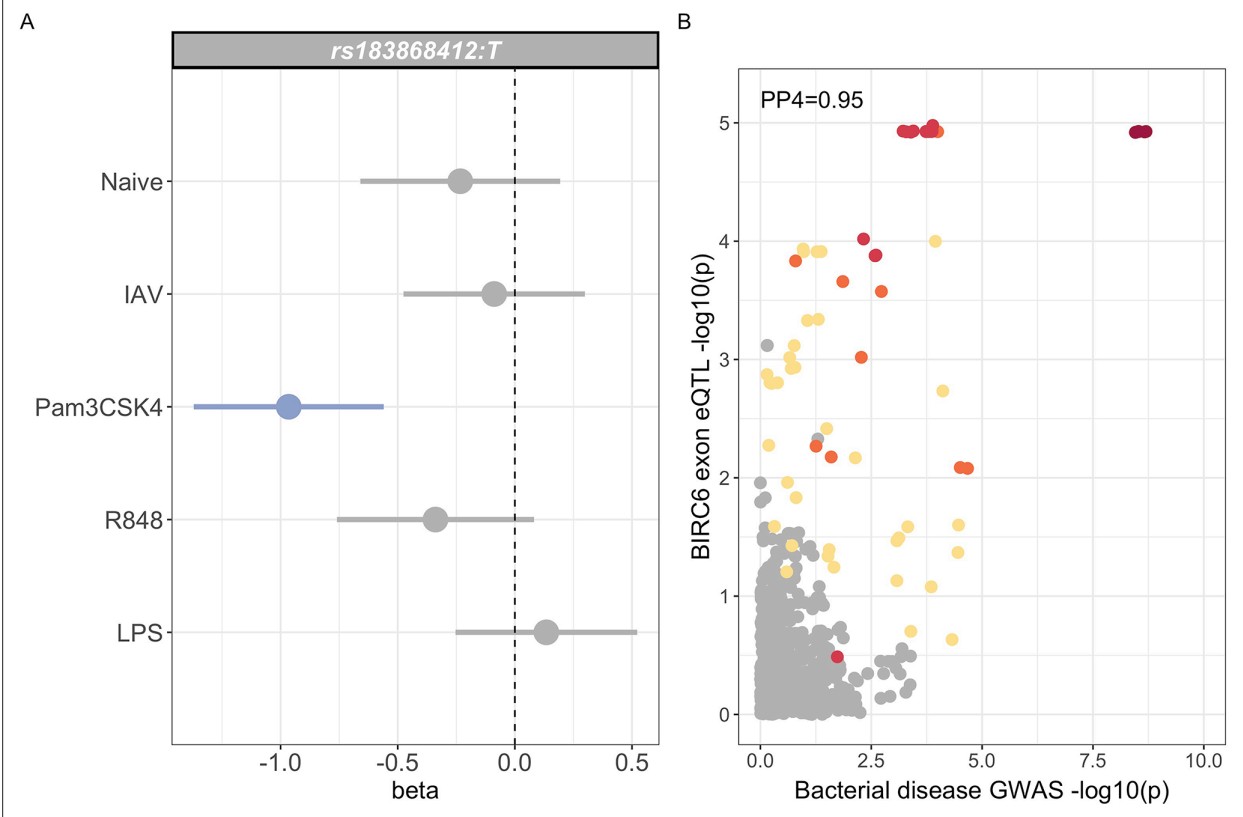

**Figure 6.** Regulatory function of rs183868412 monocytes. (**A**) Betas and 95% confidence intervals of rs183868412 association with expression of a 12 bp *BIRC6* exonic sequence (chr2:32,453,943–32,453,954) in monocytes. Monocytes are naive or stimulated with LPS (lipopolysaccharide), IAV (influenza A virus), Pam3CSK4, and R848. (**B**) Colocalisation of the *BIRC6* exon eQTL in Pam3CSK4-stimulated monocytes colocalises with the risk locus for invasive bacterial disease (PP4 = 0.951). SNPs are coloured according to linkage disequilibrium to rs183868412. Effect estimates are calculated by linear regression.

The disease-associated locus modifies risk of invasive bacterial disease caused by a broad range of pathogens, including β-haemolytic streptococci, *E. coli*, nontyphoidal *Salmonella*, *Streptococcus pneumoniae*, and *S. aureus*. Moreover it modifies risk of invasive infection in both the neonatal period and in older children. Furthermore, in contrast to the rs334 *HBB* A>T mutation (*Scott et al., 2011*), rs183868412 modifies risk of invasive bacterial disease in a manner independent of malaria, with rs183868412:T carriage increasing risk of disease across a period of falling malaria transmission and in children with and without concurrent parasitaemia.

We further demonstrate that rs183868412 mediates risk of invasive bacterial disease through the modification of *BIRC6* splicing in Pam3CSK4-stimulated monocytes. *BIRC6* (Baculovirus inhibitor of apoptosis protein repeat containing 6), also known as *BRUCE* (BIR repeat containing ubiquitin-conjugating enzyme), encodes a large member of the inhibitor of apoptosis protein (IAP) family (*Hauser et al., 1998*). Members of the IAP family bind to cognate caspases, inhibiting their activity, and thereby cell death, through occlusion of their active site (*Verhagen et al., 2001*). A proportion of IAPs also contain an E3 ubiquitin ligase allowing both direct inhibition of caspases and targeting of caspases for proteasomal degradation (*Verhagen et al., 2001*). *BIRC6* contains both inhibitor of apoptosis domains and an E2/E3 ubiquitin ligase, which function to inhibit apoptosis in response to a variety of stimuli, both by interaction with and degradation of caspase-9, but also through the ubiquitination and degradation of SMAC, an IAP antagonist (*Hao et al., 2004*; *Bartke et al., 2004*). *BIRC6* also regulates autophagosome-lysosome fusion (*Ebner et al., 2018*), and ubiquitinates (and targets for degradation) LC3, a key effector of autophagosome formation. Thus, *BIRC6* also acts as a negative regulator of autophagy (*Jia and Bonifacino, 2019*).

It is highly plausible that *BIRC6* could determine susceptibility to invasive infection through either its regulation of apoptosis or autophagy (or both). Sepsis induces marked changes in apoptosis

across a range of immune cells (*Hotchkiss and Nicholson, 2006*). There is markedly enhanced apoptosis in both dendritic cells (*Hotchkiss et al., 2002*) and in lymphocytes. Enhanced lymphocyte apoptosis is most striking in B cells and CD4+ T cells (*Hotchkiss et al., 2001*) which, at least in part, is mediated by caspase-9. The consequent lymphopoenia is correlated with both severity of sepsis and outcome (*Le Tulzo et al., 2002*). In addition to the direct effects of immune cell loss on the innate and adaptive immune responses to invasive infection, sepsis-induced apoptosis induces immune cell dysfunction, phagocytosis of apoptotic cells resulting in reduction in pro-inflammatory cytokine production, and cross-presentation of antigen from apoptotic cells to adaptive immune cells (*Albert, 2004*). In keeping with a role for regulators of apoptosis in the pathogenesis of sepsis, members of the IAP family, including *NAIP/BIRC1* and *BIRC3*, are downregulated in immune cells in patients with sepsis, as is the *BIRC6* ubiquitination target *SMAC* (*Hoogerwerf et al., 2010*). Autophagy contributes to the direct removal of intracellular pathogens and, through the degradation of invading organisms in autophagosomes, directs antigen presentation and pro-inflammatory cytokine secretion (*Deretic et al., 2013*). As above, *BIRC6* regulates autophagy through its interaction with LC3, and overexpression of *LC3B* limits inflammation and tissue injury in a mouse model of sepsis (*Lo et al., 2013*).

In keeping with a role for *BIRC6* in autophagy and apoptosis in sepsis, our data identify a role for genetic variation at *BIRC6* in determining risk of invasive infection secondary to a broad range of bacteria. This is in contrast to previously published data describing susceptibility to invasive bacterial infection, which has highlighted a prominent role for genetic risk factors that are specific to single pathogens (*Davila et al., 2010*; *Gilchrist et al., 2018*; *Rautanen et al., 2016*). In this study, the derived allele (T) at rs183868412 was associated with increased risk of bacteraemia secondary to gram-positive (β-haemolytic streptococci, *S. pneumoniae,* and *S. aureus*) and gram-negative (*E. coli*, nontyphoidal *Salmonella*) pathogens, including intracellular and extracellular bacteria, and enteric and respiratory pathogens. Moreover, rs183868412 modified risk of bacteraemia in both the neonatal period, when infection is likely to be maternally derived, and in older children, when sources of community acquired infection will be more diverse. This modulation of invasive bacterial disease risk, despite diverse sources and routes of infection and diverse mechanisms of invasion, suggests a mechanism in which genetic variation at *BIRC6* modifies risk of invasive infection downstream of initial mechanisms of infection and invasion. In an interesting parallel to this, common genetic variation at another ubiquitin-conjugating enzyme, *UBE2U*, has been shown to modify outcome in meningitis secondary to diverse pathogens in individuals of European ancestry (*Lees et al., 2019*).

Our study has some important limitations. The African-specific nature of the trait-associated variation identified here limits our ability to comprehensively interrogate the effect of that variation in immune cells. The eQTL mapping data that we utilise here (*Quach et al., 2016*) is limited in that it allows us only to consider a single immune cell type. It is also important to note that there are relatively few African-ancestry individuals in the eQTL mapping data we utilise here, and the splicing signal at *BIRC6* is based on only 10 individuals heterozygous for rs183868412:T. A more complete understanding of the role played by genetic variation at *BIRC6* plays in the pathogenesis of sepsis in African children will require more detailed expression and functional studies in African populations. In addition, there is a paucity of large-scale genetic association studies performed in African populations. The African-specific nature of the variation identified in our study therefore limits our ability to explore pleiotropic effects at this locus. Allied to this, the low MAF of rs183868412 will result in very limited power to detect selection events at the locus. Larger and richer datasets detailing genetic variation in African populations will be required to explore the wider phenotypic consequences of variation at *BIRC6*. Our study uses plasma *Pf*HRP2 concentrations to help identify children at low risk of 'true' severe malaria. Given the increasing frequency of *pfhrp2* and *pfhrp3* deletions in many settings, including in Africa (*Agaba et al., 2019*; *Feleke et al., 2021*; *Gamboa et al., 2010*), an understanding of the local prevalence of *pfhrp2/3* deletions will be important in considering how to translate this model to other settings.

Taken together, our data identify a role for *BIRC6* in the pathogenesis of invasive bacterial infections in Kenyan children. By maximising our available sample size to include children with a high likelihood of invasive bacterial infection, but without culture-confirmed infection, we facilitate novel variant discovery and reveal a common genetic architecture of invasive bacterial disease secondary to diverse pathogens. In doing so, we expand our understanding of the biology of invasive infection in African

children. In particular, these data inform our understanding of the biology shared by diverse bacterial infections causing a common clinical syndrome: sepsis.

## Materials and methods

### Study samples

Recruitment of the severe malaria cases, bacteraemia cases, and healthy controls have been described in detail elsewhere (*Ndila et al., 2018*; *Rautanen et al., 2016*). In brief, children under 14 years admitted to the high dependency ward of Kilifi County Hospital with a clinical diagnosis of severe malaria, defined as *P. falciparum* parasites on blood film and at least one of reduced Blantyre Coma Score, severe anaemia (Hb < 50 g/L), evidence of respiratory distress, hypoglycaemia, or hyperparasitaemia were eligible for recruitment as cases of severe malaria. During the study period, all children admitted to hospital, with the exception of elective surgical admissions and minor trauma, had a blood sample taken for bacterial culture (BACTEC 9050 instrument, Becton Dickinson, Franklin Lakes, NJ). Children under 14 years in whom a pathogenic organism was identified in blood were eligible for study inclusion (*Bacillus* species, coryneform bacteria, coagulase-negative staphylococci, *Staphylococcus saprophyticus,* and Viridans group streptococci were considered contaminants). Control children were recruited between 3 and 12 months of age from consecutive live births from the population which Kilifi County Hospital serves. Control children have been subject to longitudinal follow-up. Following explanation of the study, written informed consent was obtained from the parent or guardian of each child included in the study. Ethical approval was obtained from the Kenya Medical Research Institute (KEMRI) National Scientific Steering and Research Committees (approval numbers: SCC1192 and SCC967) and the Oxford Tropical Research Ethics Committee (OxTREC, approval numbers: 020–06 and 014–01).

### Models to define the probability of 'true' severe malaria

Among cases of severe malaria recruited to the study, we used probabilistic models to assign a probability of that child's clinical presentation being mediated by parasite sequestration, that is 'true' severe malaria, as described previously (*Watson et al., 2021a*; *Watson et al., 2021b*). Where available (n=1400), we used platelet counts and *Pf*HRP2 concentrations to derive the probabilistic model. In cases where *Pf*HRP2 concentration was not measured (n=800), we used white blood cell counts and platelet counts as input data to the model.

For both models (Model 1: *Pf*HRP2/platelet counts; Model 2: platelet counts/white blood cell counts), the probabilities of 'true' severe malaria were derived by fitting parametric latent class models. These assumed that each patient had a binary latent state (true severe malaria versus not severe malaria). For Model 1, we assumed that in each latent state the data were distributed as a single bivariate normal distribution. For Model 2, the data did not fit well to a two-component bivariate normal (white blood cell counts have much weaker diagnostic value) so we assumed that the data had bivariate Student's t-distribution for the severe malaria state, and a flexible mixture of bivariates for the not severe malaria state.

### Genotyping and imputation

For the discovery analysis, we utilised genotypes generated as part of GWASs of severe malaria (*Band, 2019*) and bacteraemia (*Rautanen et al., 2016*) previously reported in this population. Bacteraemia cases and controls were genotyped using the Affymetrix SNP 6.0 array and the severe malaria samples using the Illumina Omni 2.5M platform. SNP and sample quality control for both datasets are highly analogous, and have been described previously (*Band, 2019*; *Rautanen et al., 2016*). In brief, MalariaGEN SNP QC excluded poorly genotyped SNPs using the following metrics; SNP missingness >2.5%, MAF <1%, Hardy-Weinberg equilibrium (HWE) $p < 1 \times 10^{-20}$, plate effect $p < 1 \times 10^{-3}$, and a recall test quantifying changes in genotype following a re-clustering process $p < 1 \times 10^{-6}$. For Affymetrix-genotyped samples, SNP QC excluded poorly performing SNPs using the following metrics: SNP missingness > 2%, MAF < 1%, genotype probability (info) < 0.975, plate effect $p < 1 \times 10^{-6}$, and HWE $p < 1 \times 10^{-20}$. Sample QC on both platforms excluded sample outliers with respect to channel intensity, missingness, heterozygosity, population outliers, and duplicated samples (relatedness coefficient >0.75). In addition, for Affymetrix-genotyped samples, samples were

excluded in cases of discordant sex as recorded in the clinical record and imputed from mean intensities from X and Y chromosome markers.

To facilitate combining datasets, we applied an additional set of cross-platform QC procedures. We defined a shared subset of SNPs genotyped and passing SNP QC on both platforms (n=167,108), observing high levels of genotype concordance (median concordance 0.993, *Figure 3—figure supplement 4*) among the subset of samples genotyped on both platforms (n=1365). We used this shared SNP set to compute relatedness estimates and PCs in PLINK v1.90 (*Chang et al., 2015*). The major six PCs of shared genotypes differentiate self-reported ethnicity (*Figure 3—figure supplement 1*) and are non-differential with respect to genotyping platform (*Figure 3—figure supplement 1*). To harmonise QC across both platforms, we excluded MalariaGEN samples with discordant clinical and genetic sex (n=136). We further excluded one of duplicate or related sample pairs (relatedness coefficient >0.2) across platforms, retaining case samples where possible and excluding equal numbers of control samples genotyped on each platform (n=1973). Following QC, genotypes were phased using SHAPEIT2 (*Delaneau et al., 2011*), and untyped genotypes imputed genome-wide using IMPUTE2 (v2.3.2) (*Howie et al., 2011*; *Howie et al., 2009*) with 1000 Genomes Phase III as a reference panel. Following imputation, we excluded SNPs with imputation info scores < 0.5, MAF < 1% and HWE $p < 1 \times 10^{-5}$, applying each threshold for each platform and overall. Following SNP and sample QC, 14,010,600 autosomal SNPs and indels from 5400 samples (1445 bacteraemia cases, 1143 severe malaria cases, and 2812 control samples: 917 Illumina genotyped and 1895 Affymetrix genotyped) were taken forward as a combined discovery dataset for association analysis. Following QC and association analysis, we identified a further set of population outliers using ABERRANT (*Bellenguez et al., 2012*) for downstream sensitivity analysis.

## Estimation of SNP heritability

To estimate SNP heritability of bacteraemia in this population, we used genome-wide genotyping data from culture-confirmed bacteraemia cases and healthy controls genotyped with the Affymetrix SNP 6.0 array. For this analysis we used directly typed markers passing quality control as described above (n = 783,094). In addition to the sample quality control applied above, we additionally excluded one individual from each sample pair with relatedness coefficient >0.05, leaving a final sample size of 2559 (1042 cases, 1517 controls). We estimated SNP heritability using GCTA-GREML (*Yang et al., 2011*). For transformation of the heritability estimate to the liability scale, we assumed a population prevalence for bacteraemia of 2% in this setting.

## Association analysis and fine mapping

In the discovery analysis, we tested for association between genotype at each variant passing QC and invasive bacterial disease by logistic regression in an additive linear model. We used weighted logistic regression to reflect the probability of each case sample being a 'true' case of invasive bacterial infection. Cases of culture-confirmed bacteraemia were assigned a weight of 1, whereas cases of severe malaria were assigned weights of 1 – P(SM|Data), re-weighting the contribution of a case to the log-likelihood according to its probability of representing invasive bacterial infection. Control samples were assigned a weight of 1. Our regression thus assumes that the lower the probability of 'true' severe malaria, the greater the probability that a case represents culture-negative invasive bacterial disease. To control for confounding variation, we included the six major PCs of genotyping data and genotyping platform as covariates in the model. Weighted logistic regression was implemented using the *glm* function in R. As described previously (*Watson et al., 2021a*), standard errors were transformed to reflect the reduced effective sample size resulting from inclusion of sample weights in the model. We considered $p < 5 \times 10^{-8}$ to be significant.

We used a Bayesian approach to identify a set of SNPs with 95% probability of containing the causal variant at the trait-associated locus. Approximate Bayes factors (*Wakefield, 2009*) were calculated for each SNP in the region (a 200 kb surrounding rs183868412) with a prior distribution of $N(0, 0.2^2)$. All SNPs were considered equally likely to be the causal variant a priori. A set of SNPs with 95% probability of containing the causal SNP was defined as the smallest number of SNPs for which the summed posterior probabilities exceed 0.95.

## Replication samples and analysis

To replicate our findings from the discovery analysis, we used a second sample set, recruited from the same population as the discovery samples. Replication case samples were cases of bacteraemia only, and did not include cases of severe malaria without culture-confirmed bacterial infection. Case samples were recruited between 1 August 1998 and 30 October 2010. As for the discovery case samples, children under 14 years with a bacterial pathogen isolated from blood on admission to hospital were eligible for recruitment to the study. As above, control samples were recruited as part of a birth cohort from the same population, with children recruited between the ages of 3 and 12 months. Genotyping and QC procedures for these samples have been described previously. In brief, study samples were genotyped using the Immunochip Consortium (*Cortes and Brown, 2011*) array (Illumina). Sample QC was performed as for the discovery samples (above), with duplicate control samples (samples common to MalariaGEN and Immunochip controls, n=78) being removed from the replication set. As above, relatedness estimates and PCs were computed in PLINK v1.90 (*Chang et al., 2015*; *Figure 4—figure supplement 1*). SNP QC excluded the following variants: SNP missingness > 1%, MAF < 1%, and HWE $p < 1 \times 10^{-10}$. Following QC, 143,000 genotyped variants in 434 cases and 1258 control samples were taken forward for imputation. As above, imputation was performed with SHAPEIT2 (*Delaneau et al., 2011*) and IMPUTE2 (v2.3.2) (*Howie et al., 2011*; *Howie et al., 2009*) with 1000 Genomes Phase III as a reference panel.

Following imputation, we further excluded poorly imputed (imputation info score < 0.5) and rare (MAF < 1%) variants and variants with HWE $p < 1 \times 10^{-10}$. At variants associated with invasive bacterial disease ($p < 5 \times 10^{-8}$) in the discovery analysis, we tested for association with bacteraemia case-control status using logistic regression in an additive model in SNPTEST v2.5.6 (*Marchini et al., 2007*). To exclude confounding variation, we included the major six PCs of genotyping data in the model. We considered evidence of association with bacteraemia in the replication samples with $p < 0.05$ with the same direction of effect as in the discovery analysis to be significant.

## Bayesian comparison of models of association

At the locus of interest, we used multinomial logistic regression, implemented in SNPTEST v2.5.6 (*Marchini et al., 2007*) to estimate the additive effect of genotype on risk of bacteraemia stratified by pathogen, bacteraemia in the neonatal and non-neonatal periods, bacteraemia with and without *P. falciparum* parasitaemia, and bacteraemia presenting at different time periods across a period of declining malaria transmission intensity. For these analyses we used only samples with culture-confirmed bacteraemia. In each case we used control status as the baseline stratum, and included the six major PCs of genotyping data to control for confounding variation as above.

For the pathogen-stratified analysis, we defined eight case strata among the discovery cases, one for each of the seven most commonly isolated organisms (*Acinetobacter*, n=118; β-haemolytic streptococci, n=130; *E. coli*, n=141; *Haemophilus influenzae* type b, n=113; nontyphoidal *Salmonella*, n=159; pneumococci, n=390; *S. aureus*, n=152) and one stratum for the remaining other organisms (n=242). For the neonatal/non-neonatal disease analysis, we stratified cases as presenting in the first 28 days of life (n=195) or beyond that (n=1245). For the analysis comparing bacteraemia with and without malaria, we stratified cases with (n=204) and without (n=1236) *P. falciparum* on their admission blood film. For each of these analysis, case strata were compared to Affymetrix-genotyped discovery control samples (n=1895) as a baseline stratum.

For the analysis stratified across year of admission, we defined case strata by grouping into six time periods according to their date of admission: 1998–2000 (n=498), 2001–2002 (n=349), 2003–2004 (n=467), 2005–2006 (n=310), 2007–2008 (n=237), 2009–2010 (n=111). For this analysis we used both discovery (Affymetrix) and replication (Immunochip) case and control samples. This allowed better coverage of the years later in the study, which were underrepresented in the discovery samples (the discovery median admission year is 2003, cf 2005 for the replication samples). In that analysis we used multinomial logistic regression in each cohort to estimate stratum-specific effects, combining these results in a fixed effects meta-analysis using BINGWA (*Malaria Genomic Epidemiology Network et al., 2015*).

We then compared models of association using a Bayesian approach (*Rautanen et al., 2016*), considering the following models:

"Null": effect size = i.e. no association with bacteraemia.

"Same": effect size $N(0, 0.2^2)$ and fixed across strata.

Additional models consider each possible combination of a fixed effect size for associated strata and no association at other strata. For each model, we calculated approximate Bayes factors (*Wakefield, 2009*) and posterior probabilities, assuming each model to be equally likely a priori. Statistical analysis was performed in R.

## eQTL mapping and colocalisation analysis

We used the eQTL Catalogue (*Kerimov et al., 2021*) mapping pipeline (https://github.com/eQTL-Catalogue/, *Kerimov, 2022*) to map eQTL in naive and stimulated monocytes (*Quach et al., 2016*). These data include bulk RNA-seq and genome-wide genotyping data from naive and stimulated primary monocytes isolated from individuals of European (n=100) and African (n=100) ancestry (*Quach et al., 2016*). Given the African-specific nature of variation at rs183868412, we performed eQTL mapping in this dataset restricting our analysis to samples of African ancestry. The eQTL Catalogue mapping pipeline has been described previously (*Kerimov et al., 2021*). In brief, sample genotypes (Illumina HumanOmni5-Quad genotyped) were pre-phased with Eagle v.2.4.1 (*Loh et al., 2016*) and genotypes imputed with Minimac4 v.1.0.2 (*Das et al., 2016*) using 1000 Genomes Phase III as a reference panel. Gene expression, exon expression, transcript usage, and transcriptional event usage were quantified from RNA-seq data using HISAT2 v.2.1.0 (*Kim et al., 2019*), DEXSeq v.1.18.4 (*Anders et al., 2012*), and Salmon v.0.13.1 (*Patro et al., 2017*). Nominal mapping in cis was performed for each phenotype for variants within a 1 Mb window of the start of each gene using FastQTL (*Ongen et al., 2016*), including six PCs of genotyping and phenotype data as covariates in the model.

We then used the R package coloc v5.1.0 (*Giambartolomei et al., 2014*) to identify evidence of causal variants shared by our bacterial disease-associated locus of interest and regulatory genetic variation identified in our eQTL mapping data. Coloc adopts a Bayesian approach to compare evidence for independent or shared association signals for two traits at a given genetic locus. We used the `coloc.susie()` command to allow colocalisation of multiple independent signals at a single locus for each trait. We considered evidence for colocalisation for each gene and exon within a 250 kb window of the peak association (rs183868412). We considered a posterior probability >0.8 supporting a shared causal locus to be significant.

## Acknowledgements

This publication uses genotyping data from the MalariaGEN consortium project, as described in Malaria Genomic Epidemiology Network, et al. Nature Communications, 2019 (https://doi.org/10.1038/s41467-019-13480-z). This study makes use of data generated by the Wellcome Trust Case Control Consortium 2 project (Grant Reference 085475/B/08/Z). JJG and AJM are funded by National Institute for Health Research (NIHR) Clinical Lectureships. TNW and JAGS are supported by Senior Research Fellowships from the Wellcome Trust (202,800, and 098532, respectively). JAW is a Sir Henry Dale Fellow funded by the Wellcome Trust (223253/Z/21/Z). SMAART (Severe Malaria Africa – A consortium for Research and Trials) is funded by a Wellcome Collaborative Award in Science grant (209265/Z/17/Z) held in part by KM and TNW. During this work, AVSH was supported by a Wellcome Trust Senior Investigator Award (HCUZZ0) and by a European Research Council advanced grant (294557). The research was supported by the Wellcome Trust Core Award Grant Number 203141/Z/16/Z with additional support from the NIHR Oxford BRC. The views expressed are those of the author(s) and not necessarily those of the NHS, the NIHR, or the Department of Health. This research was funded by The Wellcome Trust. A CC BY or equivalent licence is applied to the author accepted manuscript arising from this submission, in accordance with the grant's open access conditions. This paper is published with the permission of the Director of KEMRI.

## Additional information

### Funding

| Funder | Grant reference number | Author |
|---|---|---|
| Wellcome Trust | 202800 | Thomas N Williams |
| Wellcome Trust | 098532 | J Anthony G Scott |
| National Institute for Health Research | | James J Gilchrist<br>Alexander J Mentzer |
| Wellcome Trust | 223253/Z/21/Z | James A Watson |
| Wellcome Trust | 209265/Z/17/Z | Kathryn Maitland<br>Thomas N Williams |
| Wellcome Trust | HCUZZ0 | Adrian VS Hill |
| European Research Council | 294557 | Adrian VS Hill |

The funders had no role in study design, data collection and interpretation, or the decision to submit the work for publication. For the purpose of Open Access, the authors have applied a CC BY public copyright license to any Author Accepted Manuscript version arising from this submission.

### Author contributions

James J Gilchrist, Conceptualization, Data curation, Formal analysis, Investigation, Visualization, Writing – original draft; Silvia N Kariuki, Data curation, Formal analysis; James A Watson, Conceptualization, Resources, Formal analysis, Writing - review and editing; Gavin Band, Conceptualization, Resources, Data curation, Writing - review and editing; Sophie Uyoga, Carolyne M Ndila, Resources, Data curation; Neema Mturi, Salim Mwarumba, Shebe Mohammed, Moses Mosobo, Data curation; Kaur Alasoo, Resources, Formal analysis, Validation; Kirk A Rockett, Conceptualization, Resources, Data curation; Alexander J Mentzer, Resources; Dominic P Kwiatkowski, Conceptualization, Resources, Supervision, Funding acquisition; Adrian VS Hill, Conceptualization, Supervision, Funding acquisition; Kathryn Maitland, Data curation, Supervision, Funding acquisition, Writing - review and editing; J Anthony G Scott, Conceptualization, Data curation, Supervision, Funding acquisition, Writing - review and editing; Thomas N Williams, Conceptualization, Data curation, Formal analysis, Supervision, Funding acquisition, Writing – original draft, Writing - review and editing

### Author ORCIDs

James J Gilchrist http://orcid.org/0000-0003-2045-6788
James A Watson http://orcid.org/0000-0001-5524-0325
Kaur Alasoo http://orcid.org/0000-0002-1761-8881
Kathryn Maitland http://orcid.org/0000-0002-0007-0645
J Anthony G Scott http://orcid.org/0000-0001-7533-5006
Thomas N Williams http://orcid.org/0000-0003-4456-2382

### Ethics

Human subjects: Following explanation of the study, written informed consent was obtained from the parent or guardian of each child included in the study. Ethical approval was obtained from the Kenya Medical Research Institute (KEMRI) National Scientific Steering and Research Committees (approval numbers; SCC1192 and SCC967) and the Oxford Tropical Research Ethics Committee (OxTREC, approval numbers; 020-06 and 014-01).

### Decision letter and Author response

Decision letter https://doi.org/10.7554/eLife.77461.sa1
Author response https://doi.org/10.7554/eLife.77461.sa2

# Additional files

## Supplementary files
• Transparent reporting form

## Data availability
Patient level genotype and phenotype data are available via the European Genome-Phenome Archive, with accession codes EGAD00010000950 (WTCCC2: bacteraemia cases and controls) and EGAD00010000904 (MalariaGEN Consortium: severe malaria cases and controls). Full GWAS summary statistics have been deposited with the GWAS Catalog with accession code GCST90094632. Code and source data underlying each figure (and figure supplement) are available at: https://github.com/jjgilchrist/Kenya_bacteraemia_malaria (copy archived at swh:1:rev:327d28a3803b92502fb58ee0bbf8ac199a10836e).

The following dataset was generated:

| Author(s) | Year | Dataset title | Dataset URL | Database and Identifier |
|---|---|---|---|---|
| Gilchrist JJ, Kariuki S, Watson JA, Band G, Uyoga S, Ndila CM, Mturi N, Mwarumba S, Mohammed S, Mosobo M, Rockett KA, Mentzer AJ, Kwiatkowski DP, Hill AVS, Maitland K, Scott JAG, Williams TN | 2022 | BIRC6 modifies risk of invasive bacterial infection in Kenyan children | https://www.ebi.ac.uk/gwas/search?query=GCST90094632 | EBI, GCST90094632 |

The following previously published datasets were used:

| Author(s) | Year | Dataset title | Dataset URL | Database and Identifier |
|---|---|---|---|---|
| Kenyan Bacteraemia Study Group, Wellcome Trust Case Control Consortium 2 (WTCCC2), Rautanen A, Pirinen M, Mills TC, Rockett KA, Strange A, Ndungu AW, Naranbhai V, Gilchrist JJ, Bellenguez C, Freeman C, Band G, Bumpstead SJ, Edkins S, Giannoulatou E, Gray E, Dronov S, Hunt SE, Langford C, Pearson RD, Su Z, Vukcevic D, Macharia AW, Uyoga S, Ndila C, Mturi N, Njuguna P, Mohammed S, Berkley JA, Mwangi I, Mwarumba S, Kitsao BS, Lowe BS, Morpeth SC, Khandwalla I, Kilifi Bacteraemia Surveillance Group, Blackwell JM, Bramon E, Brown MA, Casas JP, Corvin A, Duncanson A, Jankowski J, Markus HS, Mathew CG, Palmer CNA, Plomin R, Sawcer SJ, Trembath RC, Viswanathan AC, Wood NW, Deloukas P, Peltonen L, Williams TN, Scott JAG, Chapman SJ, Donnelly P, Hill AVS, Spencer CCA. | 2016 | WTCCC2 Bacteraemia Susceptibility (BS) samples | https://ega-archive.org/studies/EGAS00001001756 | ega, EGAD00010000950 |
| Malaria Genomic Epidemiology Network | 2015 | Genome-wide study of resistance to severe malaria in eleven worldwide populations | https://ega-archive.org/studies/EGAS00001001311 | ega, EGAD00010000904 |

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
