## [Editor Report]

Gilchrist et al. present evidence that a genetic variant in the BIRC6 gene increases the risk of invasive bacterial infection. This paper will be of interest to researchers working in areas relating to invasive bacterial infections, malaria, sepsis, and immunogenetics. The authors find evidence that the risk allele affects the splicing of BIRC6 in stimulated monocytes and that this may explain the association signal they found. Further investigations in African ancestry samples will be needed to elucidate the mechanism.

---

## [Decision Letter]

**Decision letter after peer review:**

Thank you for submitting your article "*BIRC6* modifies risk of invasive bacterial infection in Kenyan children" for consideration by *eLife*. Your article has been reviewed by 3 peer reviewers, including Alexander Young as the Reviewing Editor and Reviewer #1, and the evaluation has been overseen by Bavesh Kana as the Senior Editor. The following individual involved in review of your submission has agreed to reveal their identity: Diederik van de Beek (Reviewer #3).

Essential revisions:

The manuscript describes the identification of a novel risk locus for bacteraemia. The risk locus identified is biologically interesting and the reviewers generally thought that the methods used were appropriate and conclusions supported by the data and analyses. However, the reviewers raised some points that need addressing:

1) Please answer reviewer #1's questions pertaining to population stratification

2) In relation to reviewer #3's comments, it would help readers of the manuscript to include a figure/table giving an overview of the cohorts used in the manuscript: what data was collected and when.

3) Provide a more detailed justification for using model 2 over model 1, or provide sensitivity analyses comparing results using the two models.

4) Comment on whether there is evidence for selection on the risk variant

Please also address other comments related to improving clarity and presentation.

*Reviewer #1 (Recommendations for the authors):*

I have some questions about the controls for the population structure that were implemented.

Looking at Figure 2 Supplements 2 and 3, it looks to me like PC1 is driven by the presence of ~20 individuals with 'other' ethnicity genotyped on Affymetrix. Do the authors know what is atypical about the ancestry of these samples? Could the authors perform a sensitivity analysis where individuals with outlying values on PC1 are removed from the discovery sample?

Why were 6 principal components used for the discovery sample but only 4 for the replication sample?

To give further confidence that the main result is not a result of population stratification, it would be helpful to see a table that gives the case-control counts as well as the counts of the rs183868412 alleles stratified by self-reported ethnicity. Related to this, is there any evidence for differentiation between ethnic groups on rs183868412?

Line 322: 'excluded SNPs outside the following QC thresholds'. I found the language here very confusing, and is 'genotype probability (info) >0.975' the wrong way around?

Line 336: what is IBD>0.2? Do you mean relatedness coefficient or kinship coefficient or something else? How many individuals were excluded based on this threshold?

Line 399: 'the major principal components' -- please specify the number

For the analysis of eQTL data from Quach et al., were both European and African ancestries used? I found this somewhat unclear. What was the count of individuals with the rs183868412:T allele? And were any of them non-African? If both European and African ancestry individuals were used, I would suggest performing an equivalent analysis using only African ancestry individuals to avoid potential confounding with ancestry.

*Reviewer #2 (Recommendations for the authors):*

It seems surprising that a genotype with such an obviously deleterious effect can reach moderate frequency across multiple African populations. A comment on this seems appropriate. Might there be some favorable effect of the phenotype? Is there any evidence of recent selection? A full investigation of this question Obviously goes beyond the scope of this work but it should obviously help in interpreting the phenotype.

*Reviewer #3 (Recommendations for the authors):*

1. The authors make good arguments for lumping up data of malaria and bacteremia cases but it still feels a bit artificial. Please present sensitivity analyses for culture proven bacteremia cases only vs. controls (so leaving out the malaria cohort).

2. Weighted analysis of malaria cases. This is an interesting approach. Differences between "model 1" and "model 2" should be better explained. Also, just stating we used model 1 for down analyses and model 2 for all other cases is clearly insufficient. Please provide data in more detail to back this up. Alternatively, provide sensitivity analyses using both models – if the outcome is robust using either model 1 and 2 will provide similar results. Please show and discuss.

3. Heritability. The authors state in the intro that "..suggests that a significant portion of this variability is heritable." Could the authors include an analysis of human SNP heritability for bacteremia in the manuscript?

---

## [Author Response]

Essential revisions:The manuscript describes the identification of a novel risk locus for bacteraemia. The risk locus identified is biologically interesting and the reviewers generally thought that the methods used were appropriate and conclusions supported by the data and analyses. However, the reviewers raised some points that need addressing:1) Please answer reviewer #1's questions pertaining to population stratification.

We now perform sensitivity analysis excluding the additional population outliers identified by reviewer #1, and present the association analysis at rs183868412 stratified by self-reported ethnicity (Table 4). The association at rs183868412 is unchanged by additional outlier exclusion, and the effect estimates seen in the two major sub-populations (Giriama and Chonyi) are consistent with that seen in the overall analysis.

2) In relation to reviewer #3's comments, it would help readers of the manuscript to include a figure/table giving an overview of the cohorts used in the manuscript: what data was collected and when.

We now include this information as a new Figure (2).

3) Provide a more detailed justification for using model 2 over model 1, or provide sensitivity analyses comparing results using the two models.

We now present sensitivity analysis, restricting malaria case samples to children for whom we have data to calculate both models. In that analysis the observed association at rs183868412 is robust to the choice of model (see Figure 3-supplement 3).

4) Comment on whether there is evidence for selection on the risk variant.

We now expand on our reporting of within-Africa differentiation at the locus, reporting evidence for directional selection in 1000 Genomes Project African ancestry populations at the locus and examining evidence for pleiotropy at the locus with cardiometabolic traits in large-scale GWAS in African ancestry populations. We find no evidence for selection or pleiotropy at the locus and outline the limitations of these analyses in the discussion.

Reviewer #1 (Recommendations for the authors):I have some questions about the controls for the population structure that were implemented.Looking at Figure 2 Supplements 2 and 3, it looks to me like PC1 is driven by the presence of ~20 individuals with 'other' ethnicity genotyped on Affymetrix. Do the authors know what is atypical about the ancestry of these samples? Could the authors perform a sensitivity analysis where individuals with outlying values on PC1 are removed from the discovery sample?

We now include the following sensitivity analysis (lines 144-153):

“To address the possibility that the observed association at this locus is driven by confounding secondary to population structure, we used ABERRANT (Bellenguez et al., 2012) to define a set of outlier samples on the first two PCs of genotyping data (n=22, Figure 3—figure supplement 2). These individuals are all genotyped on the Illumina Omni 2.5M array and are both cases and controls (15 and 7 respectively). While they are predominantly individuals with less common self-reported ethnicities in our study population (19 of 22 are not Giriama, Chonyi or Kauma), they are not representative of a single self-reported ethnicity (the most common single ethnicity in this group is Digo, n=7). Excluding these samples from the association analysis at rs183868412:T did not significantly alter the association with invasive bacterial infection (p = 2.38 × 10−8, OR=2.05, 95% CI 1.59-2.64).”

Why were 6 principal components used for the discovery sample but only 4 for the replication sample?

We had used 4 principal components in the replication analysis primarily for convenience: these had been computed as part of the original WTCCC2 GWAS of bacteraemia, and the samples used for replication here were identical. To be consistent with the discovery analysis we have recomputed the principal components for these samples and updated our replication analysis to include the first 6 PCs. The effect estimate (OR=2.85, CI 1.54-5.28) and significance (p=8.90x10-4) of replication is in essence unaltered. We have updated Figure 4-supplement 1 with plots of the first 6 principal components and updated the replication and meta-analysis statistics throughout (Table 5, Figures 4A and 4C).

To give further confidence that the main result is not a result of population stratification, it would be helpful to see a table that gives the case-control counts as well as the counts of the rs183868412 alleles stratified by self-reported ethnicity. Related to this, is there any evidence for differentiation between ethnic groups on rs183868412?

We have included these data in a new Table 4. There is no evidence for differentiation between the study subpopulations. We have added the following to the text (lines 153-166):

“We further estimated the effect of rs183868412:T on invasive bacterial disease risk in four subpopulations defined by self-reported ethnicity (Giriama, n = 2,501; Chonyi, n = 1,560; Kauma, n = 472; Other, n = 384). Within each subpopulation we tested for association between genotype and case status in a weighted logistic regression model, including platform as a categorial covariate (Table 4). The minor allele frequency at rs183868412 ranged from 0.016 (Giriama) to 0.037 (Kauma), with no evidence of differentiation between subpopulations (FST = 0.001). We observed consistent effect sizes in both of the major study subpopulations; Giriama (OR=1.97, 95% CI 1.30-3.01, p = 0.0015) and Chonyi (OR=2.18, 95% CI 1.34-3.54, p = 0.0017) samples (Table 4), which together make up 83% of the study samples. Genotype at rs183868412 was also associated with less common self-reported ethnicities grouped together (OR=2.46, 95% CI 1.01-5.96, p = 0.047). Genotype was not associated with invasive bacterial disease risk in the Kauma subpopulation, however the sample size in the stratum is very limited (154 cases, 318 controls) and may simply reflect insufficient power to detect an association.”

Line 322: 'excluded SNPs outside the following QC thresholds'. I found the language here very confusing, and is 'genotype probability (info) >0.975' the wrong way around?

Apologies that this was not clear. We have reworded this sentence. You are also right the genotype probability exclusion is the wrong way round – this has been corrected.

Line 336: what is IBD>0.2? Do you mean relatedness coefficient or kinship coefficient or something else? How many individuals were excluded based on this threshold?

Apologies that this also wasn’t clear. By IBD we meant relatedness coefficient. This has been changed throughout.

Line 399: 'the major principal components' -- please specify the number

We have corrected this.

For the analysis of eQTL data from Quach et al., were both European and African ancestries used? I found this somewhat unclear. What was the count of individuals with the rs183868412:T allele? And were any of them non-African? If both European and African ancestry individuals were used, I would suggest performing an equivalent analysis using only African ancestry individuals to avoid potential confounding with ancestry.

Many thanks for this. The available mapping data from eQTL Catalogue included both European and African individuals. We have now re-mapped eQTL in that dataset restricting the analysis to individuals of African ancestry (n=100). As previously, the only evidence of colocalization with our risk locus is with an eQTL for a 12bp exonic sequence in BIRC6 (β = -0.97, p=1.18x10^-5^). We have updated Figure 6 and our results to reflect these new mapping data (lines 240-252) and have amended the methods to reflect this (lines 516-530).

Our eQTL mapping and colocalization data is based on 10 heterozygotes (MAF 5%) in the eQTL study. We now include this information in the Results section (line 247) and have added the following to the discussion (lines 333-339):

“The eQTL mapping data that we utilise here (Quach et al., 2016) is limited in that it allows us only to consider a single immune cell type. It is also important to note that there are relatively few African-ancestry individuals in the eQTL mapping data we utilise here, and the splicing signal at BIRC6 is based on only 10 individuals heterogygous for rs183868412:T. A more complete understanding of the role played by genetic variation at BIRC6 plays in the pathogenesis of sepsis in African children will require more detailed expression and functional studies in African populations.”

Reviewer #2 (Recommendations for the authors):It seems surprising that a genotype with such an obviously deleterious effect can reach moderate frequency across multiple African populations. A comment on this seems appropriate. Might there be some favorable effect of the phenotype? Is there any evidence of recent selection? A full investigation of this question Obviously goes beyond the scope of this work but it should obviously help in interpreting the phenotype.

We very much agree. To address this we have added the following to the manuscript (lines 217-237):

“Common genetic variation associated with a two-fold increased risk of invasive bacterial infection in children, in particular across a broad range of pathogens, will be subject to considerable negative selection pressure. This raises the question of whether this polymorphism might be associated with a positive advantage with respect to another life-threatening illness. The derived allele rs183868412:T, associated with increased risk of invasive bacterial disease in Kenyan children, is absent in non-African populations (https://gnomad.broadinstitute.org). Within Africa, rs183868412:T is present in all 9 African populations included in the MalariaGEN consortium project (Band et al., 2019) (Table 6), minor allele frequencies ranging from 0.011 in The Gambia to 0.034 in Malawi. There is no evidence for within-Africa differentiation at rs183868412 (p = 0.601) providing no support for a selective sweep at the locus. We further evaluated evidence for recent directional selection pressure, examining integrate haplotype scores (iHS) (Voight et al., 2006) in 7 African populations included in the 1000 Genomes Project. In those data, there is no evidence to support recent selection at the locus (minimum rank p = 0.07, maximum iHS = 1.3). Finally, to understand whether variation at this locus may be maintained in populations through beneficial effects on other traits, we examined evidence for pleiotropy at rs183868412 in a meta-analysis of GWAS data (Gurdasani et al., 2019) of 33 cardio-metabolic traits from Uganda, Ghana, Nigeria, South Africa and Kenya (sample size ranging from 2,671 to 14,126 individuals). In keeping with the MalariaGEN consortium project data, minor allele frequencies at rs183868412 range from 0.015 to 0.028. In these data there is no evidence for pleiotropy at the locus, with no evidence for an effect of rs183868412 on any of the 33 traits tested (minimum meta-analysis p = 0.078). The explanation for the persistence of this polymorphism, therefore, remains an open question”

It is noteworthy, however, that the modest allele frequencies of the candidate risk variants make any assessment of selection signature at the locus underpowered. We also now address this in the discussion (lines 342-343).

Reviewer #3 (Recommendations for the authors):1. The authors make good arguments for lumping up data of malaria and bacteremia cases but it still feels a bit artificial. Please present sensitivity analyses for culture proven bacteremia cases only vs. controls (so leaving out the malaria cohort).

We now present this on lines 175-178:

“Moreover, restricting our analysis to cases of culture-confirmed bacteraemia, the effect estimate for bacteraemia risk observed in the discovery analysis (1,445 cases, 2,812 controls; OR=2.12, 95% CI 1.60-2.82, p = 1.97 × 10−7) is consistent with that seen in the main model.”

2. Weighted analysis of malaria cases. This is an interesting approach. Differences between "model 1" and "model 2" should be better explained. Also, just stating we used model 1 for down analyses and model 2 for all other cases is clearly insufficient. Please provide data in more detail to back this up. Alternatively, provide sensitivity analyses using both models – if the outcome is robust using either model 1 and 2 will provide similar results. Please show and discuss.

We have performed sensitivity analysis as suggested. The association is robust to the choice of model. This is detailed on lines 167-175 and shown in Figure 3-supplement 3:

“To assess whether our analysis could be affected by our choice of model to define severe malaria case weights, we restricted our analysis to samples with data available to calculate estimates for P(SM|data) using both Model 1 and Model 2 (n=909 severe malaria cases). We recalculated effect estimates for the rs183868412 association with invasive bacterial disease using each model alone. The association with invasive bacterial disease at rs183868412:T is robust to the choice of the model for case weights, with effect estimates derived using Model 1 alone (OR=2.13, 95% CI 1.65-2.75, p = 6.92 × 10−9) and Model 2 alone (OR=2.05, 95% CI 1.59-2.65, p = 2.63 × 10−8) being entirely consistent with those seen in the main analysis (Figure 3—figure supplement 3).”

3. Heritability. The authors state in the intro that "..suggests that a significant portion of this variability is heritable." Could the authors include an analysis of human SNP heritability for bacteremia in the manuscript?

Many thanks for this suggestion. Using GCTA we derive an estimate for SNP heritability of 19% (95% CI 3-35%) for bacteraemia in this population. We report this estimate in lines 116-120, and detail the methods in lines 426-435.